# Crisis-driven digitalization and academic success across disciplines

**Dina Tinjić** [1] *, **Anna Nordén** [2]

**1** Department of Economics, Swedish University of Agricultural Sciences, Uppsala, Sweden, **2** Jönköping International Business School, Jönköping University, Jönköping, Sweden

* dina.tinjic@slu.se

**Data Availability Statement:** All relevant data are within the paper and its Supporting information files.

**Funding:** The authors received no specific funding for this work.

## Abstract

While the rapid digitalization in higher education, accelerated by the COVID-19 pan- demic, has restructured the landscape of teaching and learning, a comprehensive under- standing of its implications on students' academic outcomes across various academic disciplines remains unexplored. This study, therefore, aims to fill this gap by providing an in-depth examination of the effects of crisis-driven digitalization on student performance, specifically the shift to emergency remote education during the COVID-19 crisis. Lever- aging a panel dataset encompassing 82,694 individual student course grades over a span of six years, we explore the effects of digitalization across nationalities, educational levels, genders, and cru- cially, academic disciplines. Our findings are threefold: (i) firstly, we note that crisis-driven digitalization significantly impacted students' chances of passing a course and achieving higher course grades in comparison to the pre-crisis period. (ii) Secondly, we found the effect to be heterogeneous across disciplines. Notably, practical disciplines, such as nurs- ing, experienced a negative impact from this sudden shift, in contrast to more theoretical dis- ciplines such as business administration or mathematics, which saw a positive effect. (iii) Lastly, our results highlight significant variations in the impact based on educational levels and nationalities. Master's students had a harder time adapting to the digital shift than their bachelor counterparts, while international students faced greater challenges in less interna- tional academic environments. These insights underscore the need for strategic interven- tions tailored to maximize the potential of digital learning across all disciplines and student demographics. The study aims to guide educators and policymakers in creating robust digi- tal learning environments that promote equitable outcomes and enhance students' learning experiences in the digital age.

## 1. Introduction

Over the past decade, higher education has witnessed a significant shift towards digital tools, with edtech investments skyrocketing and projections suggesting the online education market could reach $350 billion by 2025 [1–4]. This change was further accelerated by the COVID-19 pandemic, which made e-learning a necessity rather than an option. As a result, scholarly arti- cles exploring the ramifications of digitalization on higher education have surged [1, 2, 5–8], highlighting the growing academic discourse surrounding this topic.

**Competing interests:** The authors have declared that no competing interests exist.

Review studies are crucial to synthesize these diverse perspectives into a more holistic understanding of e-learning in higher education. M. A. Fauzi's bibliometric study [1] serves as a notable contribution, offering a comprehensive overview of recent advancements and spotlighting the most influential publications. These unprecedented times have required rapid adjustments in educational strategies. In the face of the pandemic's challenges, educators have been forced to swiftly integrate digital technologies, rework teaching methods, and modify assessment strategies. The recent rise in innovative online and blended learning models [9, 10], coupled with advancements in big data analytics, virtual/augmented reality, and AI [4, 5, 11–13], is evidence of this accelerated evolution.

Furthermore, over the years, scholars have consistently identified the need to adapt educational techniques to optimize learning outcomes [14]. A growing body of research provides insights for educators navigating the complexities of innovative pedagogical methods. For example, the theory of gamified learning [15, 16], AI literacy model [12], increasing student engagement using educational technologies [14], and students' feedback literacy [17], all provide valuable practical guidance for understanding, adopting, and applying innovative teaching practices. Recent studies have also proposed digital identity models for data security in e-learning [18], as well as a barrier diagnostic framework [19] aimed to assist in identifying and addressing challenges arising from employment of active students' learning practices. While the body of research is expansive and continues to grow, there remains a pressing need for better alignment between research findings and teaching practices [19]. Notably, with educational technologies, students' behavioral engagement stood out the most, followed by cognitive dimensions [14]. Moreover, research suggests that innovative teaching techniques, such as game-based learning can enhance student interest and knowledge retention [16]. The studies accentuate the obstacles faced by instructors, including the extensive time required for course adaptation amidst tight schedules, technical barriers, or skepticism regarding proposed methodologies. As higher education contemplates a digital transition, research suggests that equipping and supporting educators is vital. Without structured training and consistent technical and pedagogical support, the practicality of a fully digitally optimized higher education system may be called into question. While the existing literature has extensively documented the tools, techniques, and challenges of digital education, a significant gap persists in understanding the direct implications of these rapid shifts on student performance across various academic and course disciplines. The ongoing debate regarding the efficacy of online learning compared to traditional face-to-face instruction further highlights this knowledge gap, signaling the urgent need for a deeper exploration of how digitalization impacts student academic outcomes, considering a range of academic disciplines and diverse student characteristics. Given the transformations in the higher educational landscape, as highlighted earlier, it becomes crucial to study whether these changes are truly fostering academic excellence or inadvertently creating disparities in student outcomes. This paper seeks to bridge that gap by exploring which disciplines experienced the most pronounced positive or negative shifts due to crisis-driven digitalization and pinpoint the student groups most sensitive to these changes. By doing so, it seeks to offer insights for developing strategies that are tailored to leverage the full potential of digital tools across a spectrum of academic fields and student demographics, thereby enhancing the resilience and adaptability of the education system in facing potential future disruptions.

## 1.1 Background literature

The evidence on students' academic success and the factors affecting the performance gap between online and face-to-face course settings is mixed and mostly inconclusive [20–36].

Certain studies indicate better student performance in an online course setting, with students reporting a deeper understanding of course material, improved communication with teaching staff, and higher overall engagement and satisfaction [21, 25, 37, 38].

Conversely, several studies favor conventional face-to-face learning [23, 30, 32, 39–42], suggesting that traditional learning models yield somewhat superior student outcomes [30, 32, 35, 40], but no significant changes in attendance, dropout rates, or the time spent on assigned tasks [32]. In terms of students' perceptions of the two settings, the results are varied [20, 22, 43, 44]. Students prefer different components of the two learning models, with flexibility being a favoured characteristics of online courses, while the social aspect and active learning facilitated through group and class discussions are seen as major advantages of face-to-face courses. These factors can potentially affect the perceived quality in each learning model.

Further evidence suggests that performance gaps between online and traditional classroom settings may be more pronounced in certain academic disciplines, particularly in social sciences and applied professions [21, 45]. Practical-related courses, however, have shown a positive response to the use of digital learning platforms, such as Zoom and Moodle, during the COVID-19 pandemic, with nearly 40% of students reporting improved academic achievement [44]. While online degree programs have broadened access to higher education, their graduates' employment outcomes have, nevertheless, raised concerns about program quality. Candidates with traditional degrees tend to receive more callbacks from employers than their counterparts with online degrees [46, 47]. This issue has become even more salient due to the pandemic induced shift to online education. Recent research has indicated that while students' academic performance may have improved post-pandemic, their work-readiness appears to have decreased in comparison to pre-pandemic students [48]. However, previous research on online higher education, conducted before the pandemic, might have leaned towards an overly pessimistic outlook.

This perspective is largely influenced by two key limitations: endogeneity bias and dependence on outdated instructional technology. It's suggested that data obtained from the pandemic-driven shift to online instruction could offer a more accurate depiction of the potential benefits and challenges associated with online higher education [49]. Notably, the sudden shift to e-learning serves as a natural experiment, allowing us to bypass selection bias prevalent in education research [29], and analyze the causal effect of crisis-driven digitalization on student academic performance. Selection bias refers to the systematic differences between those who are selected for study and those who are not (see for example [34, 49–52]). In the context of education research, selection bias may manifest when students self-select into different modes of learning, such as online versus traditional classrooms, often based on factors that may also affect their academic performance. This bias can cloud the true effect of the mode of learning on student performance.

However, in the context of the pandemic, the move to online learning was a top-down decision made by educational institutions, and not a choice available to students. This unique scenario allows us to circumvent selection bias and analyze the true causal effect of crisis-driven digitalization on student academic performance. Moreover, in the field of higher education research, the decision to utilize qualitative or quantitative measures typically hinges on the specific objectives of the study. Quantitative data, in particular, remains underutilized in empirical studies investigating the effects of digitalization on academic success. For our research design, we've been influenced by a handful of studies that have effectively used quantitative data and employed identification strategies similar to ours [27, 29, 40, 41]. To measure this effect, we use students' course grades, which poses a potential limitation to our study. The reliability of the measure (i.e. the measurement error) mostly depends on the difference between an individual's observed outcome and the true outcome in an assessment [53, 54]. However, the lack

of a proper comparable measurement for students' learning and course outcomes is old news for the field [24, 53–61].

## 1.2 Study objectives

The variability in the results provided by the available literature further signifies how challenging it is to deliver the same course-instruction quality in both settings while mitigating the negative effects that arise when designing online versions of the course assessments. Our study aims to navigate the multidimensional impacts of crisis-driven digitalization on higher education, with three specific objectives:

Firstly, we seek to determine whether crisis-driven digitalization has influenced student academic success and how this effect varies across schools. The analysis measures this effect through the probability of passing a course, as well as through course grades, which allows for a more precise evaluation of the effect.

Secondly, our study elucidates how the ramifications of crisis-driven digitalization diverge between different student cohorts. This includes differences among (i) master's versus bachelor's students, and (ii) international versus Swedish students, with further distinction between (iii) tuition-paying and tuition-exempt students (considering students from the European Economic Area (EEA) are exempted from paying tuition in Sweden).

Finally, we delve into the heterogeneous influences of digitalization across different academic disciplines. We exploit individual-level data to categorize courses from all four schools by discipline, helping to identify areas with the most pronounced positive or negative effects.

In navigating these objectives, we address the research gap concerning the impacts of crisis-driven digitalization on different academic disciplines and student characteristics, a pressing issue in this post-Covid-19 period. Our research examines the complexities and nuances surrounding the "digital push" in higher education, highlighting the importance of understanding the variable implications of online learning models across different disciplines. The results reveal which disciplines had the largest positive and negative effects from crisis-driven digitalization and identify the most vulnerable student groups. We conclude by discussing potential sources of these heterogeneities and providing directions for future research.

Our argument, grounded in literature, proposes the level of practical elements in a discipline as a significant determinant of the direction, magnitude, and significance of crisis-driven digitalization effects. This proposition is corroborated by a clear pattern in our findings, which suggest that more practical disciplines—such as nursing—experience a more negative impact from crisis-driven digitalization compared to more theoretical disciplines—like business administration or math.

While the uniqueness of our dataset and the granularity of our analysis are notable strengths, we recognize the limitation of the study being centred on a single Swedish university. Therefore, we encourage future research to replicate this study with a more representative sample of the Swedish student population to further validate and expand upon our findings.

## 2. Materials and methods

### 2.1 Data description

To assess the impact of crisis-driven digitalization on students' academic success in higher education, we utilize a rich dataset from a Swedish university for six academic years, from the fall of 2014 to the summer of 2020. During the treatment period (March 2020 to June 2020), the university pivoted completely to remote education due to COVID-19 pandemic restrictions but had traditionally relied on face-to-face teaching and in-person examinations preceding the pandemic. The university consists of four schools, focusing on health/social work,

education/communication, engineering, and business/economics, which represent broader categories of academic disciplines. The data used are based on the full population of students at specified periods.

The effects of crisis-driven digitalization differed slightly across different schools, where some practical courses consisting of fieldwork that could not be replaced by digital solutions were postponed or cancelled and therefore excluded from our sample. Moreover, the small number of digital courses conducted remotely before the COVID-19 crisis period were eliminated from our sample to better isolate the digitalization effect. Last, the sample also excluded data on exchange students due to their short stay (less than a year), which made it impossible to follow them over time. In addition, the majority of incoming and outgoing student exchanges were cancelled during the period used in the study, meaning that course grade observations for exchange students were collected only during the control period of the study.

At this university, the semesters are divided into two autumn and two spring quarters and students take 15 credits (two 7.5-credit courses or one 15-credit course) each quarter. The government recommendation to switch to remote teaching came into effect in the first spring quarter of 2020. More specifically, spring 1 implemented face-to-face teaching but online exams, while in spring 2, both classes and examinations were conducted remotely. Therefore, all courses attended before the start of the first spring quarter in 2020 comprise the control group, while courses taken during the second spring quarter in 2020 fall into the treatment group.

We focused on students with at least one course grade obtained before and one after the crisis-driven switch to digital teaching models, i.e., courses that started after March 17, 2020 (when all higher education institutions in Sweden implemented digital teaching models, following the government recommendation), which corresponded to our control and treatment groups, respectively. In this way, we decreased the sample bias and focused only on students whom we could follow in both periods. During the treatment period, all teaching and examinations were moved completely online, meaning that the dataset contained no observations that had a hybrid learning variant during crisis-driven digitalization.

This gave us a dataset of 6,182 program students with a total of 82,694 individual course grade observations across four schools and 1,122 courses ranging across 35 disciplines (which we categorized by reviewing each course syllabus). That includes 1,260 health/social work students, 1,722 education/communication students, 1,590 business/economics students, and 1,801 engineering students. The sum of the students in each academic field is greater than 6,182 because some students take elective courses at other schools and thus are counted twice. The data included information on student ID, course name and ID, discipline, individual course grades, course start date and course end date (i.e., final exam date), the date the student started the program, level of the program (bachelor's or master's), gender (male or female), and nationality. The data on students' nationalities allowed us to distinguish among (1) EEA students (who come from either Sweden or another country within the EEA) and (2) non-EEA students (who come from countries outside the EEA). Another important difference between the two groups was the relative cost of their education. In Sweden, EEA students are exempt from tuition fees, as everything is covered by the state's welfare system, while non-EEA students are required to cover tuition fees. Next, if we instead categorize the groups into national students (specifically those holding a Swedish passport) and (2) international students (non-Swedish passport), we can explore whether being away from home and family during the pandemic affected students' abilities to perform academically. We do this to investigate whether international students represent the more vulnerable group of the two. Paying full tuition fees for digital versions of courses, limited social interactions, and distance from home

and family during a global health crisis can be very frustrating for international students. It was therefore reasonable to assume that this would be confirmed in our results section.

The data does not include information on examiners or instructors. However, if the course assessments, such as the learning outcomes, the core literature, or the grading criteria, changed, the course received a new course name and course ID. Thus, even though we could not control for instructor effects, major within-course changes were controlled for by including the course ID. During the COVID-19 crisis period in our data (March 17–June 14, 2020), the course IDs did not change even though all courses implemented remote education and all examinations were conducted remotely. Our data did not allow us to distinguish between the crisis-driven digitalization effect on course pass rates and dropout rates since the dropout rates were registered as failing grades.

## 2.2 Empirical strategy

To measure students' academic success, we used (i) course success rate and (ii) course grade. Course success rate is a binary variable that took a value of 1 if the individual passed the course on the first attempt and 0 otherwise. We assigned a passing grade only when the individual passed the course examination on the first attempt. This was to ensure that we captured the crisis-driven digitalization effects and not the effects of various retake opportunities (the university offers multiple retake opportunities throughout the academic year, meaning that a student who started a course in e.g., 2017 had many retake opportunities relative to our treatment group, which included students who started a course in March 2020). The course grades varied across the four schools due to differences in grading systems. The schools of health/social work and business/economics use a grading scale of F–A, which we converted to a 0–5 scale (thus: A(5) = at least 90% of the maximum score in the course; B(4) = 80%; C(3) = 70%; D(2) = 60%; E(1) = 50%; and F(0)<50%). The schools of education/communication and engineering use a scale of fail, pass, and pass with distinction, which we converted to a 0–2 scale. Students who did not pass the course assessment requirements on the first attempt failed and received a grade of 0. How grades are determined varies across disciplines. For instance, in business administration courses, student grades are based on a combination of test scores and instructor assessments of efforts such as group projects, while most economics and statistics courses base the grades solely on final exam scores. Courses in the areas of nursing, social work, and biomedicine often combine final exam scores on theory and work-based elements through collaboration with health and social institutions. Similarly, courses in the area of pedagogy and teaching in primary and preschool education, as well as media science, rely heavily on the combination of theory and practice, including internships, training, and work-based experience. In mathematics and engineering, course grades are based on a mix of theory, application, and individual and group projects. Most engineering programs also require work experience, where credits are earned through a co-op or internship. Each academic year is divided into quarters, including two autumn quarters (weeks 35–42; and weeks 43–50) and two spring quarters (weeks 3–11; and weeks 13–21). During one academic year, a full-time program student takes eight courses, with two courses being taken at a time. Hence, to control for any time effects of the COVID-19 restriction period (i.e., March–June 2020), we included time-specific variables, such as the academic year (i.e., 2019/2020), the study year in which the student was registered at the time of taking the course (first, second, third, etc.), which ideally should correspond with the student's academic year of study, as well as the semester in which the course was taken and the grade received (i.e., fall dummy equal to 1 if the course was taken in the fall quarter and 0 if in the spring). Furthermore, since our identification strategy compared pass rates and grades in courses taken before and after crisis-driven digitalization, it could be that

**Table 1. Descriptive statistics by student characteristics.**

|  | All Schools | Health/ Social Work | Education/ Communication | Business/ Economics | Engineering |
|---|---|---|---|---|---|
| Pass Rate | 0.714 | 0.729 | 0.823 | 0.757 | 0.585 |
|  | (0.452) | (0.445) | (0.382) | (0.429) | (0.493) |
| Grade | - | 2.182 | 1.040 | 2.611 | 1.049 |
|  |  | (1.700) | (0.627) | (1.755) | (1.059) |
| Passing Grade † | - | 2.994 | 1.264 | 3.448 | 1.792 |
|  |  | (1.239) | (0.441) | (1.087) | (0.764) |
| CDD | 0.129 | 0.154 | 0.130 | 0.137 | 0.108 |
|  | (0.335) | (0.361) | (0.337) | (0.343) | (0.311) |
| Male | 0.432 | 0.159 | 0.226 | 0.517 | 0.680 |
|  | (0.495) | (0.366) | (0.418) | (0.500) | (0.467) |
| Master's † | 0.102 | 0.002 | 0.016 | 0.249 | 0.116 |
|  | (0.302) | (0.039) | (0.125) | (0.432) | (0.320) |
| International | 0.133 | 0.020 | 0.009 | 0.371 | 0.108 |
|  | (0.340) | (0.141) | (0.097) | (0.483) | (0.310) |
| EEA † | 0.077 | 0.019 | 0.007 | 0.259 | 0.041 |
|  | (0.267) | (0.137) | (0.080) | (0.438) | (0.198) |
| Tuition-paying † | 0.066 | 0.001 | 0.003 | 0.193 | 0.073 |
|  | (0.248) | (0.037) | (0.054) | (0.395) | (0.259) |
| N | 82,694 | 14,338 | 21,224 | 20,741 | 26,391 |

Notes: Standard deviation in parentheses. *CDD* (Crisis-driven digitalization, 1 if course after COVID-19, 0 otherwise); *male* (1 if male, 0 if female);*master's* (1 if master's, 0 if bachelor's), *international* (1 if not Swedish, 0 if Swedish); *EEA* (1 if EEA, 0 if Swedish); *tuition-paying* (1 if tuition-paying, 0 if Swedish) (International students are categorized into tuition-paying students that come from EEA countries excluding Sweden, and tuition-exempt students that come from non-EEA countries).

† The number of observations is lower for these variables.

the courses observed in the treatment period were more or less difficult than courses in the other periods. To control for this possibility, we included course fixed effects (FE) through course ID dummies. Moreover, to explore heterogeneity in the effect on students' academic success, we created interaction terms between the treatment (crisis-driven digitalization) and academic level (bachelor's or master's), gender (male or female), and nationality (Swedish, EEA, and non-EEA students). Data on students' nationality allowed us to further distinguish between tuition-exempt and tuition-paying students. Finally, we checked for heterogeneities across disciplines and chose three disciplines with the highest number of observations from each school (thus, 12 out of 35 disciplines are presented in the results section). Table 1 shows descriptive statistics of our variables.

Fig 1 depicts the effect of crisis-driven digitalization on a student's probability of passing a course on the first attempt. The effect is presented for each school individually. As discussed in the results section, crisis-driven digitalization negatively affected all schools except business and economics, where students were found to perform better academically. Due to large variations in the dataset, the period used in the figure includes only data from the spring semester of 2018/2019 to the spring semester of 2019/2020. To identify the impact of crisis-driven digitalization on academic success, we utilized the panel nature of our data and estimated the following two-way Fixed Effects (FE) model:

$$AcademicSuccess_{i,j} = \alpha_i + \beta_1 CDD_{i,j} + \beta_2 CourseID_j + \beta_3 X_{i,j} + \varepsilon_{i,j}$$

where *AcademicSuccess*$_{i,j}$ refers to the academic success of student i in course j measured by (i)

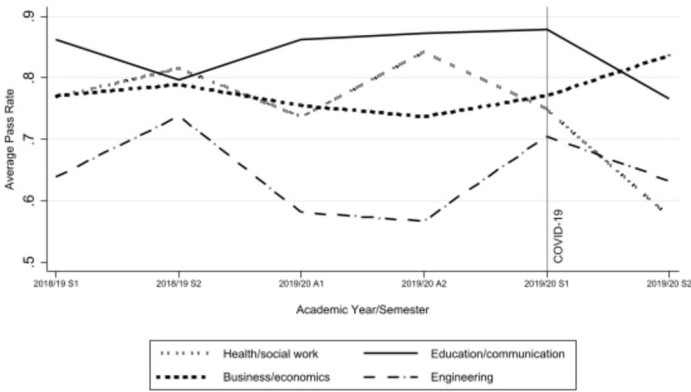

**Fig 1. Effects of crisis-driven digitalization on students' average pass rates.** Note: Tha academic year is divided into quarters (two autumn and two spring semesters). A1 (week 35–42); A2 (week 43–50); S1 (week 3–11); S2(week 13–21).

course success rate and (ii) course grade. $CDD_{i,j}$ (crisis-driven digitalization) is an explanatory dummy variable with a value of 1 if the course started after March 17, 2020, i.e., courses that were conducted entirely online and driven by the COVID-19 crisis. $CourseID_j$ controls for course FE, where each course ID is treated as a dummy variable. $X_{ij}$ refers to a vector of time-variant student-level controls (i.e., academic year, study year, and fall dummy), $\alpha_i$ is the student-level intercept (FE) capturing time-invariant student-level variables, and $\varepsilon_{ij}$ is the error term. Model (1) is estimated both for the whole university and for the four schools separately. Separating the schools allowed us to both explore the heterogeneity among academic fields and use the course grades based on each school's grading system, which allowed for more precision in the results. The model was estimated first for *all* program students and then for all program students who passed a course on the first attempt.

We also estimated the two-way FE model including variables for the interaction between $CDD_{i,j}$ and academic level (bachelor's or master's), gender, and nationality. For the academic level, we created a dummy variable ($Master_i$) with a value of 1 if a course was a master's-level course and zero if it was a bachelor's-level course. For gender, we created a dummy variable ($Male_i$) with a value of 1 if a student was male and 0 if female. To explore differences between national and international students, we created three different variables: $International_i$, which took a value of 1 if a student was non-Swedish and 0 if Swedish, $EEA_i$, which took a value of 1 if a student was from any EEA country other than Sweden and 0 if Swedish, and $Tuition_i$, which took a value of 1 if a student was from outside the EEA and therefore had to pay a tuition fee and 0 if Swedish. $EEA_i$ and $Tuition_i$ were created to identify potential differences in effects between different types of international students.

Our results are presented as follows: first, we present the effect of crisis-driven digitalization on the probability of passing a course; second, we check for the effect of crisis-driven digitalization on the grades of students who passed a course on the first attempt. This allows us both to check for the overall effect on the probability of passing for all program students and to investigate whether it was easier or harder for students to achieve a higher course grade under crisis-driven digitalization, taking into consideration the grading system of each school. Finally, we conclude the results with suggestions for future research.

## 3. Results and discussion

Table 2 presents the estimated effect of crisis-driven digitalization on students' academic success in terms of the probability of passing a course. We find that in general, for students across

**Table 2. The impact of the crisis-driven digitalization (CDD) on students' probability of passing a course.**

| | All Schools | Health/ Social work | Education/ Communication | Business/ Economics | Engineering |
|---|---|---|---|---|---|
| **CDD Impact on the prob. ff passing** | -0.006 | -0.105*** | -0.016 | 0.072*** | -0.001 |
| | (0.006) | (0.014) | (0.011) | (0.012) | (0.013) |
| Course fixed effects | Yes | Yes | Yes | Yes | Yes |
| Controls | Yes | Yes | Yes | Yes | Yes |
| Constant | 0.299*** | -0.070 | -0.054 | 0.454*** | 1.193*** |
| | (0.086) | (0.106) | (0.234) | (0.102) | (0.077) |
| $R^2$ | 0.289 | 0.379 | 0.240 | 0.120 | 0.368 |
| N | 82,694 | 14,338 | 21,224 | 20,741 | 26,391 |

Notes: Clustered (at student level) standard errors in parentheses.

*$p < 0.10$,

**$p < 0.05$,

***$p < 0.01$.

Included control variables at the student level: course ID, the academic year of the course, study year in the program when the course is taken, and semester dummy (1 if the course is taken in the fall semester, 0 otherwise).

all schools (column 1), crisis-driven digitalization did not have a significant effect on the probability of passing a course, but there were heterogeneities in terms of magnitude and direction (columns 2–5). A large negative effect was found for health/social work students, where the probability of passing a course decreased by approximately 10.5 percentage points. These results are in line with previous studies pointing to the difficulty of offering more hands-on practice and instructor-student interaction online [45], cancelling clinical rotations due to the risk of transmitting the virus, and a lack of resources, which affected a majority of students in the field of health and medicine [62–66]. In contrast, a positive impact was found for business/economics students, for whom the probability of passing a course increased by approximately 7.2 percentage points. For engineering and education/communication students, the effects were nonsignificant when controlling for course FE and other time-variant variables. The heterogeneities in the effect of crisis-driven digitalization on students' academic success across schools could be explained by differences in the measures taken by the teaching staff at each school to adjust course structures, examinations, and grading criteria when transferring courses online. The increase in students' course pass rates and grades (see Tables A8 and A9 in S1 Appendix) in business and economics, for instance, might be a result of more generous grading, which is in line with the findings of Bird et al. (2022) [42] and Bulman & Fairlie (2022) [67].

Another explanation could be that teaching practices varied across schools. For example, some instructors provided recorded lectures that were available on demand, while others offered only live lectures. Furthermore, the self-directed learning skills of students, their attitudes toward online teaching, academic dishonesty, and difficulty in the efficient delivery of online course formats might explain the heterogeneity effects across faculties [33, 34, 39, 45, 68, 69].

Next, Table 3 shows the results obtained when examining the impact of crisis-driven digitalization on grades conditional on passing a course on the first attempt. Here, we used standardized course grades for each school as the dependent variable. There was a negative effect on grades for education/communication students, i.e., a decrease of approximately 0.06 grade points (column 2), suggesting that even though crisis-driven digitalization did not influence

**Table 3. The impact of the crisis-driven digitalization (CDD) on course grades of students who pass a course (on the first attempt).**

|  | Health/ Social work | Education/ Communication | Business/ Economics | Engineering |
|---|---|---|---|---|
| **CDD Impact on** | -0.098 | -0.057*** | 0.287*** | -0.041 |
| **student grades** | (0.057) | (0.015) | (0.032) | (0.026) |
| Course fixed effects | Yes | Yes | Yes | Yes |
| Controls | Yes | Yes | Yes | Yes |
| Constant | 3.000*** | 0.810*** | 3.295*** | 3.030*** |
|  | (0.490) | (0.139) | (0.278) | (0.176) |
| $R^2$ | 0.216 | 0.164 | 0.251 | 0.351 |
| N | 10,449 | 17,461 | 15,704 | 15,450 |

Notes: Clustered (at student level) standard errors in parentheses.

*$p < 0.10$,

**$p < 0.05$,

***$p < 0.01$.

Included control variables at the student level: course ID, the academic year of the course, study year in the program when the course is taken, and semester dummy (1 if the course is taken in the fall semester, 0 otherwise).

the probability of passing a course, students who passed a course had greater difficulty in achieving higher grades under the effects of digitalization.

In contrast, business/economics students who passed a course on the first attempt had an easier time achieving higher grades during the crisis (an increase of approximately 0.29 grade points) (column 3). This means that in business and economics, it was easier both to pass a course and to achieve higher grades during the crisis-driven digitalization period in the spring of 2020. An explanation for this positive effect could be found in the results of an evaluation survey conducted by a qualitative research team at the sampled university regarding how students and teachers perceived the transition to emergency remote education during spring 2020 (details on the objective, survey design, and limitations see the research project description in Augustine et al. (2021) [70].

The questionnaire was administered through an online survey targeting students and faculty at all four schools. Overall, 1,220 students and faculty participated in the survey. The survey results were used as complementary material to help explain some of the mechanisms behind the differences we found in academic success between different disciplines. For instance, compared with the other schools, the faculty in business and economics expressed greater difficulties, including technological difficulties, in changing instructional formats from on-site to online. In other words, the business and economics faculty had a harder time than others adapting some courses to an online setting, which could have been reflected in a significant increase in course pass rates and grades. These difficulties that arose when changing the instructional formats of the courses could have further impacted the faculty's grading leniency. Additionally, the survey showed that relative to other schools, more business/economics students preferred online courses and perceived that it was very easy to transition from on-site to online. This finding indicates that these students' perceptions regarding the remote education setting were more optimistic, which in turn may have increased their motivation and self-discipline, and thus their grades, during the pandemic. For the health/social work and engineering students, we found no significant effect on grades for those passing the course on the first attempt (columns 1 and 4).

The results point to a negative effect on course pass rates for health/social work and a positive effect for business/economics (Table 2); additionally, the negative effects on the grades of

education/communication students (Table 3) were confirmed when the model was run for all program students with grades as a dependent variable (see Tables A8 and A9 in S1 Appendix).

Next, interaction terms were created to explore how the impact of crisis-driven digitalization on student academic success varied across academic levels and nationalities. Similar to the above, Table 4 shows the effect of crisis-driven digitalization on the probability of passing, while Table 5 shows the results obtained when analyzing the impact on grades conditional on passing a course on the first attempt. Given the lack of international students and students in 2-year master's programs at the schools of health/social work and education/communication, we focused on the results for the schools of business/economics and engineering.

We found that the impact of crisis-driven digitalization on pass rates for master's business/economics students was significant and negative, suggesting that master's business/economics students were less positively affected by crisis-driven digitalization than bachelor's students (Table 4 column 1). This difference between academic levels in business and economics could be attributable to several factors. For instance, bachelor's students may have received relatively more support than master's students during the transition phase, as master's students may have been perceived to be more autonomous. Furthermore, master's students may be exposed to more stress since they are closer to entering the labour market than bachelor's students. For engineering students, there was no significant difference in pass rates between master's and bachelor's students during the crisis-driven digitalization period when course FE and time-specific controls were added (Table 4 column 2).

Turning to the effects on grades for those who passed the courses (Table 5), we found that master's engineering students seemingly achieved higher grades during the crisis-driven digitalization period (column 2) than bachelor's students, who experienced a decrease in course grades, while we found no significant difference across academic levels in business and economics (column 1). When investigating heterogeneities across national and international students, we found a significant negative effect on international engineering students, whose

**Table 4. The impact of the crisis-driven digitalization (CDD) on the probability of passing a course (interactions included).**

| | Master's | (S.E.) | International | (S.E.) | EEA | (S.E.) | Tuition | (S.E.) |
|---|---|---|---|---|---|---|---|---|
| **Business/Economics** | | | | | | | | |
| CDD Impact | 0.065*** | (0.017) | 0.085*** | (0.013) | 0.093*** | (0.013) | 0.094*** | (0.014) |
| CDD*Interaction | −0.141*** | (0.035) | −0.041* | (0.017) | −0.042* | (0.020) | −0.035 | (0.025) |
| Constant | 0.481*** | (0.133) | 0.449*** | (0.102) | 0.449*** | (0.108) | 0.271 | (0.138) |
| R2 | 0.135 | | 0.120 | | 0.125 | | 0.114 | |
| N | 15, 060 | | 20, 741 | | 17, 615 | | 16, 177 | |
| **Engineering** | | | | | | | | |
| CDD Impact | 0.005 | (0.014) | 0.016 | (0.013) | 0.006 | (0.014) | 0.012 | (0.013) |
| CDD*Interaction | −0.042 | (0.036) | −0.191*** | (0.034) | −0.178*** | (0.053) | −0.207*** | (0.040) |
| Constant | 1.195*** | (0.077) | 1.195*** | (0.077) | 1.158*** | (0.074) | 1.196*** | (0.078) |
| R2 | 0.368 | | 0.369 | | 0.371 | | 0.373 | |
| N | 26, 391 | | 26, 391 | | 24, 549 | | 25, 383 | |

Notes: Clustered (at student level) standard errors in parentheses.

*$p < 0.10$,

**$p < 0.05$,

***$p < 0.01$.

Included control variables at the student level: course ID, the academic year of the course, study year in the program when the course is taken, and semester dummy (1 if the course is taken in the fall semester, 0 otherwise).

**Table 5. The impact of the crisis-driven digitization (CDD) on course grades of students who pass a course on the first attempt (interactions included).**

|  | Master's | (S.E.) | International | (S.E.) | EEA | (S.E.) | Tuition | (S.E.) |
|---|---|---|---|---|---|---|---|---|
| **Business/Economics** |  |  |  |  |  |  |  |  |
| CDD Impact | 0.306*** | (0.043) | 0.288*** | (0.034) | 0.274*** | (0.035) | 0.342*** | (0.036) |
| CDD*Interaction | −0.153 | (0.088) | −0.007 | (0.045) | −0.029 | (0.053) | 0.015 | (0.062) |
| Constant | 3.086*** | (0.320) | 3.294*** | (0.279) | 3.323*** | (0.343) | 2.846*** | (0.373) |
| *R2* | 0.273 |  | 0.251 |  | 0.257 |  | 0.256 |  |
| **N** | 11, 148 |  | 15, 704 |  | 13, 920 |  | 12, 215 |  |
| **Engineering** |  |  |  |  |  |  |  |  |
| CDD Impact | −0.092*** | (0.027) | −0.047 | (0.026) | −0.062* | (0.026) | −0.053* | (0.026) |
| CDD*Interaction | 0.461*** | (0.082) | 0.090 | (0.069) | 0.046 | (0.092) | 0.086 | (0.087) |
| Constant | 3.003*** | (0.173) | 3.030*** | (0.176) | 3.168*** | (0.177) | 3.021*** | (0.176) |
| *R2* | 0.352 |  | 0.351 |  | 0.357 |  | 0.345 |  |
| **N** | 15, 450 |  | 15, 450 |  | 14, 415 |  | 14, 816 |  |

Notes: Clustered (at student level) standard errors in parentheses.

*$p < 0.10$,

**$p < 0.05$,

***$p < 0.01$.

Included control variables at the student level: course ID, the academic year of the course, study year in the program when the course is taken, and semester dummy (1 if the course is taken in the fall semester, 0 otherwise).

probability of passing a course decreased by approximately 19 percentage points (Table 4, column 2) compared to Swedish students. For business/economics, we found that the increased probability of passing a course was lower for international students than for Swedish students (column 1).

Dividing international students into EEA (i.e., tuition-exempt students) and non-EEA (i.e., tuition-paying students) allowed a deeper analysis of heterogeneities across different types of international students. We found that for international business/economics students, the decreased positive effect of crisis-driven digitalization was motivated by the lower performance of EEA students (Table 4, column 1). For engineering, on the other hand, the significant negative effect of crisis-driven digitalization on the probability of passing a course was found to occur among both EEA and non-EEA students (Table 4 column 2). Table 5 shows that there was no significant effect on the grades of international students compared to Swedish students for those passing the course. It is important to consider that the business/economics school is highly international, with 43% of the students being non-Swedish and 13% being tuition-paying students, while engineering has only approximately 10% international students. Thus, the impact was seemingly more severe for students in less international environments. One explanation could be that more international environments may be better prepared to provide different types of solutions, which may make international students feel more included and hence more motivated. It is also possible that international students returned home once the classes moved online, which could also have affected their academic performance. These possibilities, however, could not be confirmed by the data. We also checked for performance heterogeneities across genders but found no significant results for any of the schools except engineering, where male engineering students were more negatively affected by crisis-driven digitalization than female engineering students (with an approximately 6 percentage points decrease in the probability of passing a course). The results table for gender heterogeneities is available upon request.

**Table 6. The impact of the crisis-driven digitalization (CDD) on the probability of passing a course with respect to the course discipline.**

| Course discipline | CDD Impact | (S.E.) | Constant | (S.E.) | $R^2$ | N |
|---|---|---|---|---|---|---|
| **Health/social work** | | | | | | |
| *Nursing* | −0.144*** | (0.020) | 0.457** | (0.168) | 0.500 | 4, 792 |
| *Social Work* | −0.073** | (0.026) | −0.056 | (0.196) | 0.183 | 2, 574 |
| *Biomedical Lab. Science* | 0.075 | (0.041) | −0.061 | (0.132) | 0.283 | 2, 386 |
| **Education/Communication** | | | | | | |
| *Teaching* | 0.066** | (0.023) | −1.703*** | (0.426) | 0.307 | 6, 258 |
| *Pedagogy* | −0.099*** | (0.022) | −0.109 | (0.200) | 0.223 | 5, 559 |
| *Media and Communication Science* | 0.030 | (0.024) | −1.876*** | (0.263) | 0.109 | 3, 973 |
| **Business/Economics** | | | | | | |
| *Business Administration* | 0.021 | (0.015) | 0.435** | (0.144) | 0.121 | 12, 824 |
| *Economics* | 0.193*** | (0.032) | 0.460** | (0.172) | 0.096 | 4, 579 |
| *Statistics* | 0.353*** | (0.083) | 0.627* | (0.297) | 0.132 | 1, 556 |
| **Engineering** | | | | | | |
| *Civil Engineering* | 0.069* | (0.033) | 0.574** | (0.152) | 0.504 | 4, 995 |
| *Mechanical Engineering* | 0.132*** | (0.031) | 1.480*** | (0.263) | 0.388 | 3, 949 |
| *Mathematics* | −0.102** | (0.035) | 0.552** | (0.206) | 0.129 | 3, 797 |

Notes: Clustered (at student level) standard errors in parentheses.

*$p < 0.10$,

**$p < 0.05$,

***$p < 0.01$.

Included control variables at the student level: course ID, the academic year of the course, study year in the program when the course is taken, and semester dummy (1 if the course is taken in the fall semester, 0 otherwise).

Lastly, we present the results obtained when investigating how crisis-driven digitalization affected students' probability of passing a course across different disciplines. Our data cover 35 disciplines. The results for the 12 disciplines with the largest samples (three from each school) are presented in Table 6, while Table A6 in the S1 Appendix shows the heterogeneous effect on grades conditional on passing across the same disciplines.

For health/social work students, the negative effect of crisis-driven digitalization on academic performance was driven mostly by courses in nursing and social work, where the probability of passing decreased by approximately 14 and 7 percentage points, respectively (Table 6). Additionally, for students taking courses in social work, it became harder to achieve higher grades than in the pre-COVID-19 crisis period (students experienced a decrease of approximately 0.43 grade points; see Table A6 in S1 Appendix). This is in line with the literature discussed earlier that emphasizes the importance of practical skills in medicine and healthcare education as well as the challenge of carrying out courses in these fields online.

For education/communication students, there was a significant negative effect of crisis-driven digitalization for courses in pedagogy, where the probability of passing decreased by approximately 10 percentage points, whereas it increased by approximately 7 percentage points for courses in teaching (referring to preschool and elementary-level children) (Table 6). An increase in the probability of passing for students taking courses in teaching can be explained by the fact that preschools and elementary schools in Sweden did not close during the pandemic and that courses in this field rely heavily on practical experience as a learning tool.

In business and economics, the positive effect seems to have been driven mostly by economics and statistics courses, for which students' probability of passing increased during the

crisis-driven digitalization period by 19 and 35 percentage points, respectively (Table 6). This might indicate that economics and statistics courses were harder to conduct online, and the teaching staff may have had a harder time adjusting to the online examination or may have been more lenient when grading. Additionally, in business administration, economics, and statistics, students had a significantly easier time achieving higher grades upon passing the course (see Table A6 in S1 Appendix) than in the pre COVID-19 crisis period. In business administration courses, the grades of those who passed increased by approximately 0.24 grade points, while economics and statistics course grades increased by approximately 0.92 and 0.98 grade points, respectively. This can be explained either by students being more comfortable presenting their assignments and projects online and taking online tests because of less performance pressure or by a potential increase in academic dishonesty among students. In business administration courses, grades are determined by combining grades from group work, presentations, and exams, which are mostly essay style. This assessment structure also makes it harder to cheat in these courses. On the other hand, in economics and statistics, grades are solely based on final exam grades. Since the final exams are of a mathematical and problem-solving nature, it is easier for students to collaborate during exams or use "homework help" websites such as Chegg to cheat during online exams [71].

For engineering, the results were more diverse. We found a significantly positive effect for course grades in civil and mechanical engineering, where the probability of passing increased by approximately 7 and 13 percentage points, respectively, while for mathematics courses, the probability of passing decreased by approximately 10 percentage points (Table 6). Furthermore, students taking courses in the field of mechanical engineering and mathematics had a harder time achieving higher grades than in the pre-COVID-19 crisis period (Table A6 in S1 Appendix), as the grades of those who passed decreased by approximately 0.25 and 0.20 grade points, respectively. This may indicate that the teaching staff in the engineering courses decreased the threshold for passing a course but were stricter with grading. For mathematics courses, it was both harder to pass and harder to achieve a higher course grade than in the pre-COVID-19 crisis period, potentially indicating that the teaching staff were stricter when designing online versions of the courses and exams.

Our findings showed that the effect of crisis-driven digitalization on students' academic success varied across academic disciplines. We found that the probability of passing a course decreased for health/social work students and increased for business/economics students. Our results also suggested that for business/economics students who passed the course on their first attempt, it was easier to achieve higher grades than in the pre-crisis-driven digitalization period, while for education/communication students, the grades of those who passed decreased during the digitalization period. We also found that students in practically oriented disciplines (e.g., nursing and teaching) were negatively affected by digitalization in terms of passing a course, while those in more theoretically oriented disciplines (e.g., economics, statistics, and mechanical engineering) were positively affected.

Our results are in line with Bulman & Fairlie (2022) [67] and Bird et al. (2022) [42], who found increased probabilities of passing courses in spring 2020. Moreover, our finding of heterogeneity in the effects of crisis-driven digitalization across academic fields is supported by Odriozola-González et al. (2020) [72]. In addition, our positive results on academic success for business/economics students are supported by Gonzalez et al. (2020) [73].

The heterogeneity in our results is supported by the literature on the impact of online education on students' academic performance compared with the impact of traditional face-to-face education. The literature generally points to a negative difference in the performance of online relative to traditional education, a so-called performance gap, which depends largely on individual, peer, and course characteristics [25, 26, 45, 74, 75]. At an individual level, students'

online success is determined by their levels of self- directed learning, including their self-discipline, self-regulation, and cognitive capabilities [33, 34, 39, 45, 68, 69]. Given that self-directed learning skills may vary depending on gender, nationality, and educational level [45], we investigated differences in the effect of digitalization based on such factors as well.

The negative effect on the academic success of health/social work students can be explained by the finding that wider performance gaps exist in fields that depend heavily on hands-on practice and instructor-student interactions [45]. Many recent studies on the negative effect of digitalization due to the COVID-19 pandemic on students' academic success have attributed this finding to the fact that many medical and healthcare students around the world were prevented from participating in clinical rotations due to the risk of transmitting the virus as well as a lack of resources [62–66]. This could explain why we found the largest negative effect on this school, particularly for nursing students. Teacher education is another discipline in which hands-on experience and the acquisition of practical skills are important, but as traditional educational practices are being reshaped, the switch to online education is viewed here as an opportunity rather than a challenge [76]. Another potential explanation for the disparities between teacher and nursing students could be attributed to the changes in the labour market during the COVID-19 pandemic restrictions in Sweden. While schools remained open throughout the entire period, allowing practical elements to be carried out by teacher students, nursing students faced a unique challenge. They not only encountered obstacles in performing their practical training but also experienced an increased demand for their skills [77]. This combination of factors may have diverted students' attention away from their academic studies, potentially impacting their overall performance. For economics, the literature is ambiguous. Coates et al. [31] found that economics students in face-to-face sections scored almost 10–18% higher than students in online sections. This result was supported by Brown & Liedholm [39]. In contrast, Navarro & Shoemaker [38] found that online economics students performed significantly better than students in face-to-face settings. Furthermore, we found differences in student-teacher interactions and heterogeneity in the need for such interaction in different courses. Since students in face-to-face settings are required to actively engage in the learning process, their relatively better performance is attributed to the benefits of direct student-teacher interactions. The relatively poorer performance of online students is attributed to the lack of self-discipline since online students reported spending less than three hours per week on a course, while according to attendance records, students in face-to-face sections spend a minimum of three hours weekly just attending the class. Navarro & Shoemaker [38] reported similar findings that were based on surveys of online instructors and analysis of courses in more than 50 colleges offering over 100 online economics courses. Poor grades in online economics courses were found to be the result of a lack of motivation and self-direction, which, as they noted, many students find easier to generate through a web of student-to-student and professor-to-student interactions. Finally, some of our results can be explained by findings suggesting that cheating occurs more often in an online environment [71, 78]. Lancaster & Cotarlan [71] explained increased cheating during online tests by an increase in the activity of STEM students on Chegg, one of many file-sharing sites where students can cheaply and quickly purchase cheating solutions or "homework help". Such sites were found to be especially popular among students in business, computing, and accounting. More recent studies suggest that in addition to cheating, teachers' grading leniency impacted the academic success of students in terms of course withdrawal and failure rates following the switch to online studies [42, 67].

## 4. Conclusion

The digitalization of higher education offers both opportunities and challenges. We use panel data on individual student course grades over 6 years to investigate the impact of crisis-driven digitalization due to the switch to emergency remote education during the COVID-19 crisis period. Our results suggest that (i) crisis-driven digitalization significantly affected students' probability of passing a course and their ability to achieve higher course grades relative to the precrisis period; (ii) there is great heterogeneity among different academic disciplines, with health/social work students being significantly negatively affected and their business/economics counterparts being positively affected; (iii) the effect of the pandemic is highly heterogeneous across disciplines, with the largest negative effect in nursing, social work, pedagogy, and mathematics and the largest positive effect in economics and statistics courses; (iv) there are differences in academic levels, with master's students' academic success being more negatively affected during the crisis-driven digitalization period than bachelor's students; and (v) international students, both EEA (i.e., tuition-exempt students) and non-EEA (i.e., tuition-paying students), are more negatively affected by crisis-driven digitalization in less international environments. This paper contributes to the literature on digitalization by identifying the impacts of online teaching in various academic disciplines, avoiding selection bias. Our results highlight the importance of designing effective digital strategies that maximize the potential of online teaching and learning across all academic disciplines. Educators and institutions should consider the specific needs and challenges of different disciplines when implementing digital tools and pedagogies. Practical disciplines, for example, may require more hands-on and experiential learning approaches, even in an online environment. The variation in the impact of digitalization across educational levels and nationalities emphasizes the need for tailored support mechanisms. Master's students may require additional assistance and resources to adapt to online learning, while international students may benefit from targeted support to navigate challenges in less international academic environments. This customization calls for investments in competence development for educators to enhance their digital teaching skills. Educators need training and support to effectively engage students in online learning environments and overcome the disciplinary and context-specific challenges identified in this study. The unique needs of different disciplines and student populations also suggest that a one-size-fits-all approach may not be suitable. Policies should be flexible. Finally, our findings point towards the importance of preparing students for digital learning experiences. As pointed out by previous literature, students need digital literacy skills and resources to effectively engage with online coursework. Providing training and support to ensure students are equipped with the necessary tools and knowledge to succeed in a digital learning environment is crucial. By considering these implications, stakeholders can work together to optimize digital learning experiences, promote equitable outcomes across disciplines and student groups, and ensure that digitalization leads to enhanced students' learning experiences in the digital age.

### 4.1 Limitations and future research

While our study offers valuable insights into the effects of crisis-driven digitalization on students' academic success across various disciplines and student characteristics, it comes with limitations. The data sourced from a single Swedish university may limit the generalizability of our findings. It's important to consider that universities vary in their resources, student demographics, teaching practices, and digital infrastructures, all of which could impact the effects of digitalization on student success. Further, external factors such as caregiving responsibilities, financial burdens, or limited access to the internet or other digital technologies could have

influenced students' academic success during the pandemic. Acknowledging these potential confounding factors is crucial. Future research should strive to broaden the applicability of these results while maintaining depth of exploration, as previous studies often overview the general effects without considering individual academic disciplines.

Our study also uses measurable indicators of academic success, such as course pass rates and grades. However, there are many other important aspects of student learning and engagement that we did not examine, such as students' motivation, critical thinking skills, creativity, and overall well-being. Future research could explore these aspects and delve into how to better cater to the unique needs of each academic discipline during digital transitions.

Lastly, we see the necessity for more extensive review studies that focus on heterogeneities across academic disciplines. These would help synthesize the existing literature and present a more comprehensive view of the impacts of crisis-induced digitalization. Future investigations should specifically target understanding the unique challenges that different academic disciplines face during such digital transitions. By exploring targeted interventions to alleviate these challenges, we can enhance our adaptive strategies during crises. Moreover, a long-term view of digitalization's impact on students' academic performance and career trajectories is paramount. Such an analysis would be invaluable for shaping future pedagogical practices in an increasingly digital educational landscape.

## Supporting information

**S1 Data.**
(XLSX)

**S1 Appendix.**
(PDF)

## Acknowledgments

We thank the University Services for providing us with the data. We also thank Meliha Halilić, Zangin Zeebari, Jens Rommel, Luca Repetto, Kristoffer Månsson, Helena Nilsson, Emma Lappi, and Toni Duras for very useful comments, as well as Laura Villalobos for proofreading the final manuscript. Finally, we thank seminar participants at the sample university for their useful comments. We are responsible for any remaining errors.

## Author Contributions

**Conceptualization:** Dina Tinjić, Anna Nordén.

**Data curation:** Dina Tinjić, Anna Nordén.

**Formal analysis:** Dina Tinjić.

**Investigation:** Dina Tinjić.

**Methodology:** Dina Tinjić, Anna Nordén.

**Software:** Dina Tinjić, Anna Nordén.

**Supervision:** Anna Nordén.

**Writing – original draft:** Dina Tinjić, Anna Nordén.

**Writing – review & editing:** Dina Tinjić, Anna Nordén.

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
