## [Decision Letter · Decision Letter 0]

12 May 2023

PONE-D-23-00621Crisis-driven digitalization and academic success across disciplines.PLOS ONE

Dear Dr. Tinjic,

Thank you for submitting your manuscript to PLOS ONE. After careful consideration, we feel that it has merit but does not fully meet PLOS ONE’s publication criteria as it currently stands. Therefore, we invite you to submit a revised version of the manuscript that addresses the points raised during the review process.

We look forward to receiving your revised manuscript.

Kind regards,

Mukhtiar Baig, Ph.D.

Academic Editor

PLOS ONE

Additional Editor Comments:

It needs major revision.

Reviewers' comments:

Reviewer's Responses to Questions

**Comments to the Author**

1. Is the manuscript technically sound, and do the data support the conclusions?

Reviewer #1: Yes

Reviewer #2: Partly

2. Has the statistical analysis been performed appropriately and rigorously? 

Reviewer #1: I Don't Know

Reviewer #2: No

3. Have the authors made all data underlying the findings in their manuscript fully available?

Reviewer #1: No

Reviewer #2: Yes

4. Is the manuscript presented in an intelligible fashion and written in standard English?

Reviewer #1: Yes

Reviewer #2: No

5. Review Comments to the Author

Reviewer #1: Thank you for the opportunity to review this manuscript on crisis-driven digitalization and academic success. It is an interesting study that leveraged the opportunity created by the global pandemic to study academic outcomes.

While I found the study interesting and the manuscript to be generally well written, there were sections that I found confusing (which may be based on my disciplinary norms). There was significant overlap between the various sections and I frequently had to reorient myself to the headings provided. Generally research based manuscripts have distinct sections (Introduction, aim, methods (data collection, data analysis, ethics), findings, discussion (implications, limitations), conclusion). I think restructuring this paper would help with clarity.

I also encourage you to think about some of the limitations of your study and include a section that addresses these limitations. While students were certainly forced into a digital form of education, so was the rest of the world. There were various confounding factors that may have impacted students academic success that were not addressed in your study and I think they should be acknowledged (care giving responsibilities, financial burdens, limited access to internet or other digital technologies). Interesting that your findings suggest caring professionals (nursing, social work) had the largest negative effect on academic success. Might this be related to the unprecedented demands placed on these professions during the pandemic? I think a closer and more critical look at potential explanations for your findings is warranted.

Reviewer #2: Review report-

PONE-D-23-00621

Title: Crisis-driven digitalization and academic success across disciplines

General comment: The paper has investigated an important and well-time topic. Yet the paper lacks many elements. Hence a rigorous review is needed to improve this before it can be published. Having said that let me provide specific feedback for section by section.

Abstract: It has not expressed the purpose of this research. More words are used to explain research design that can be expressed precisely while the purpose, some important element of findings and implication could be the part of the abstract. This approach is missing.

Key words should be increased

Introduction: The introduction lacks of providing an explicit importance of a research problem. Also, the research gap is not identified and the scope of the paper is not cleared mapped by pointing the research aim and objectives. Hence the outing of research question is missing. The author also needs to explore how this work is going to still relevant in the post Covid-19 atmosphere. I suggest that author could read some important publications in this regard and should make a best use of them. Here is some recommended papers to look and to use them.

Does online technology provide sustainable HE or aggravate diploma disease? Evidence from Bangladesh—a comparison of conditions before and during COVID-19- DOI: 10.1016/j.techsoc.2021.101677

Can online higher education be an active agent for change? —comparison of academic success and job-readiness before and during COVID-19- DOI: 10.1016/j.techfore.2021.121008

Online technology: Sustainable higher education or diploma disease for emerging society during emergency—comparison between pre and during COVID-19- DOI: 10.1016/j.techfore.2021.121034

Access, attendance and performance in urban K8 education during pre- and post-COVID-19 restrictions in Bangladesh: comparison of students in slums, tin-sheds and flats- DOI: 10.1080/03004279.2022.2109183

The author neither make a compressive literature review to give light to a conceptual framework or to develop a theoretical underpinning. Author needs to check the impact of digitalisation on education – whether it is a complete ally or enemy or offers mixed experience. This is missing.

Research design: Naming university name is not a right fashion. It breaks the research confidently issue. Explanation on data analysis is missing. Also the data collection phase is not explained and justified.

Results seem interesting. However, results must be presented with coherence following a theme. Hence results are to be presented to answer the research questions or based on the hypothesis developed. Here too many results and discussions with a specific goal and direction.

Theoretical implication is not expressed. And there is no convincing conclusion/ concluding remark.

Language demands editing. Serious attentions are needed to be paid for technical issue such as citations, reference and formatting.

Good luck with the revision if that is given as a chance by the editor

6. PLOS authors have the option to publish the peer review history of their article (what does this mean?). If published, this will include your full peer review and any attached files.

Reviewer #1: No

Reviewer #2: No

---

## [Author Response · Author response to Decision Letter 0]

12 Oct 2023

Subject: Response to Reviewers for Manuscript ID PONE-D-23-00621, “Crisis-driven digitalization and academic success across disciplines.”

Dear Dr. Mukhtiar Baig,

We appreciate the time and effort you and the reviewers have dedicated to evaluating our manuscript. The feedback and suggestions offered have been invaluable, helping us refine and enhance our work.

We have submitted:

1) A comprehensive 'Response to Reviewers' document, addressing every comment raised during the review process.

2) A word file of the 'Revised Manuscript with Track Changes' to clearly illustrate the changes implemented.

3) The underlying data set (excel file)

4) An unmarked, clean version of this revised manuscript. 

We believe that these changes have significantly strengthened the manuscript and we look forward to hearing your thoughts on the revised version. Should further modifications be required, we are open to your guidance and feedback.

Our financial disclosure statement remains accurate and unchanged. Please let us know if additional information in this regard is needed.

Confident in the revisions made, we trust that our manuscript now aligns with PLOS ONE’s publication standards. We eagerly await your positive response.

Sincerely,

Dina Tinjić,

PhD Student

 

Response to the Editor:

1. Comment: Please ensure that your manuscript meets PLOS ONE's style requirements, including those for file naming...

Response: Thank you for highlighting the style and file naming requirements. We have revised our manuscript in line with the guidelines provided in the PLOS ONE style templates. The changes include the formatting of the main body, title, authors, and affiliations. We have also adhered to the recommended file naming conventions.

2. Comment: Please update your submission to use the PLOS LaTeX template...

Response: We appreciate your direction regarding the use of the PLOS LaTeX template. We have updated our manuscript accordingly, ensuring the LaTeX formatting adheres to the provided sample. We trust that these updates comply with PLOS ONE's standards for LaTeX submissions. We also believe this step immensely improved the readability and clarity of our manuscript, and we thank you for the comment. 

3. Comment: Specify where the minimal data set underlying the results described in your manuscript can be found...

Response: We appreciate your attention to the data availability statement. We have now included the minimal data set underlying our study, following approval from our university's statistics department. The data excel file is uploaded together with the rest of the documents. 

All identifiable information has been fully anonymized, ensuring we have adhered to ethical guidelines. There are no restrictions on sharing this data publicly. We trust this will enable the replication of the study findings and kindly request you to update our Data Availability statement accordingly.

4. Comment: We note that you have indicated that data from this study are available upon request...

Response: We appreciate your attention to the data availability details in our manuscript. In our initial submission, we inadvertently stated that data would be available upon request. However, we wish to clarify that our study's data set has now been thoroughly anonymized from the outset, in compliance with our institution's ethical guidelines and respect for our students' privacy. The data set does not contain any identifiable or sensitive information.

Given the anonymity of the data, there are no ethical or legal restrictions that prevent us from sharing it publicly. Therefore, as stated previously, we have now uploaded the minimal anonymized data set necessary to replicate our study findings. 

We apologize for any confusion our initial statement may have caused and kindly request that our Data Availability statement be updated to reflect the public availability of our data.

 

Response to Reviewer 1

1. Comment: Is the manuscript technically sound, and do the data support the conclusions?

Response: We appreciate Reviewer 1's assessment that our manuscript is technically sound and that our data supports our conclusions. We have made sure to maintain the technical rigour in our revision.

2. Comment: Has the statistical analysis been performed appropriately and rigorously?

Response: Reviewer 1 indicated uncertainty about our statistical analysis. In response to this concern, we have taken the initiative to have the statistical analysis thoroughly reviewed to ensure its validity. While we acknowledge the importance of addressing any potential limitations and improving the analysis, we would like to emphasize that our current approach, specifically employing a two-way fixed effects (FE) model, is a widely accepted method for panel data analysis in the context of natural experiments, such as the external shock of the COVID-19 pandemic. We are fully committed to addressing any shortcomings and enhancing the statistical analysis as necessary. To facilitate this process, we kindly ask Reviewer 1 to specify more clearly which parts. 

3. Comment: Have the authors made all data underlying the findings in their manuscript fully available?

Response: Reviewer 1 indicated that all data was not fully available. We have addressed this in our revision by providing access to our anonymized data set through a stable public repository, as mentioned earlier in our letter.

4. Comment: Is the manuscript presented in an intelligible fashion and written in standard English?

Response: We are grateful to Reviewer 1 for acknowledging the clarity and language of our manuscript. We have made sure to maintain this standard in our revision.

5. 

(i) Comment: While I found the study interesting and the manuscript to be generally well-written, there were sections that I found confusing. There was significant overlap between the various sections and I frequently had to reorient myself to the headings provided.

Response: We are grateful for Reviewer 1's helpful comments regarding the organization of the manuscript. Based on this feedback, we have restructured the paper to follow the suggested PLoS ONE format (1. Introduction, 2. Materials and Methods, 3. Results and Discussion, 4. Conclusion), which should enhance its clarity and coherence.

(ii) Comment: I also encourage you to think about some of the limitations of your study and include a section that addresses these limitations.

We are grateful for Reviewer 1's recommendation to further elaborate on the limitations inherent to our study. In our initial submission, we discussed the aspect of selection bias and instructor fixed effects in section 4.1, the Empirical Strategy. This section also includes a detailed explanation of the robustness of our approach in circumventing these potential issues. 

In response to your feedback, we have now added an explicit section focusing on the limitations of our study, where we acknowledge potential confounding factors that may have influenced the academic success of students during the pandemic, such as caregiving responsibilities, financial constraints, and limited access to the internet or other digital tools. We also emphasized the limited generalisability of our results and the issue of measurement error. Naturally, we also further developed our discussion on future research opportunities. This expanded section now incorporates and addresses the limitations we have identified, thereby paving the way for more comprehensive future investigations. This section aims to helps the reader understand the potential gaps in our design, and in the field.

*The following subsection was added in the manuscript text:*

4.1 Limitations and future research 

While our study offers valuable insights into the effects of crisis-driven digitalization on students' academic success across various disciplines and student characteristics, it comes with limitations. 

The data sourced from a single Swedish university may limit the generalizability of our findings. It's important to consider that universities vary in their resources, student demographics, teaching practices, and digital infrastructures, all of which could impact the effects of digitalization on student success. Further, external factors such as caregiving responsibilities, financial burdens, or limited access to the internet or other digital technologies could have influenced students' academic success during the pandemic. Acknowledging these potential confounding factors is crucial. Future research should strive to broaden the applicability of these results, while maintaining depth of exploration, as previous studies often overview the general effects without considering individual academic disciplines.

Our study also uses measurable indicators of academic success, such as course pass rates and grades. However, there are many other important aspects of student learning and engagement that we did not examine, such as students’ motivation, critical thinking skills, creativity, and overall well-being. Future research could explore these aspects and delve into how to better cater to the unique needs of each academic discipline during digital transitions. 

Lastly, we see the necessity for more extensive review studies that focus on heterogeneities across academic disciplines. These would help synthesize the existing literature and present a more comprehensive view of the impacts of crisis-induced digitalization.

(iii) Comment: Interesting that your findings suggest caring professionals (nursing, social work) had the largest negative effect on academic success. Might this be related to the unprecedented demands placed on these professions during the pandemic? I think a closer and more critical look at potential explanations for your findings is warranted.

Response: This is a valuable observation from Reviewer 1 and we agree that the increased demands on caring professionals during the pandemic could have had a significant impact on their academic success. We have now included this in the discussion section, offering potential explanations for this finding in the context of the pandemic's specific challenges. However, it is important to note that we lack data on the extent to which the students in our data experienced an increase in their labour supply during this period.

* The following text was added in the manuscript text (Results and discussion section): *

“The negative effect on the academic success of health/social work students can be explained by the finding that wider performance gaps exist in fields that depend heavily on hands-on practice and instructor-student interactions [19]. Many recent studies on the negative effect of digitalization due to the COVID-19 pandemic on students’ academic success have attributed this finding to the fact that many medical and healthcare students around the world were prevented from participating in clinical rotations due to the risk of transmitting the virus as well as a lack of resources [45–49]. This could explain why we found the largest negative effect on this school, particularly for nursing students. Teacher education is another discipline in which hands-on experience and the acquisition of practical skills are important, but as traditional educational practices are being reshaped, the switch to online education is viewed here as an opportunity rather than a challenge [59]. Another potential explanation for the disparities between teacher and nursing students could be attributed to the changes in the labor market during the COVID-19 pandemic restrictions in Sweden. While schools remained open throughout the entire period, allowing practical elements to be carried out by teacher students, nursing students faced a unique challenge. They not only encountered obstacles in performing their practical training but also experienced an increased demand for their skills (Ritva et al., 2022). This combination of factors may have diverted students' attention away from their academic studies, potentially impacting their overall performance.”

We have also included additional supporting literature:

1. Rosenbäck, Ritva, Björn Lantz, and Peter Rosén. 2022. "Hospital Staffing during the COVID-19 Pandemic in Sweden" Healthcare 10, no. 10: 2116. https://doi.org/10.3390/healthcare10102116

We believe these changes have strengthened our manuscript and addressed the concerns raised. We thank Reviewer 1 for their insightful feedback. 

Response to Reviewer 2

1. Comment: Is the manuscript technically sound, and do the data support the conclusions?

Response: Reviewer 2 partially agreed that our manuscript is technically sound and that our data supports our conclusions. 

In response to this concern, we have taken the initiative to have the statistical analysis thoroughly reviewed to ensure its validity. While we acknowledge the importance of addressing any potential limitations and improving the analysis, we would like to emphasize that our current approach, specifically employing a two-way fixed effects (FE) model, is a widely accepted method for panel data analysis in the context of natural experiments, such as the external shock of the COVID-19 pandemic. In the revised manuscript, we have made additional efforts to ensure the clarity of our analysis, discussion, and conclusions. These enhancements aim to provide a more robust and transparent description of our work.

2. Comment: Has the statistical analysis been performed appropriately and rigorously?

Response: We thank Reviewer 2 for their comment on our statistical analysis. We have revisited our analysis to ensure its rigor and appropriateness. We also reached out to our colleagues for further comments. However, as mentioned previously, employing a two-way fixed effects (FE) model is a widely accepted method for panel data analysis in the context of natural experiments, such as the external shock of the COVID-19 pandemic. We are fully committed to addressing any shortcomings and enhancing the statistical analysis as necessary, but without specific points of concern, we are somewhat uncertain about what changes would best address this concern. Thus, we kindly ask Reviewer 2 for more detailed feedback in case of any potential shortcomings in our research design. 

3. Comment: Have the authors made all data underlying the findings in their manuscript fully available?

Response: We appreciate Reviewer 2's feedback, but there may have been a misunderstanding. In our initial submission, we didn't upload our anonymized dataset to a public repository. However, in response to the journal's data policy, we've now corrected this and included the underlying data set in our resubmission documents. We also anonymized the university name to maintain confidentiality, as suggested. 

4. Comment: Is the manuscript presented in an intelligible fashion and written in standard English?

Response: We appreciate Reviewer 2's comment concerning the language and presentation of our manuscript. We would like to clarify that the manuscript underwent professional editing prior to the initial submission. As none of the authors are native English speakers, we sought the expertise of a professional language service to ensure that our work is expressed in standard and clear English.

We understand the importance of making our study accessible and easily comprehensible to a wide range of readers, and we have now closely reviewed the manuscript again to rectify any unnoticed grammatical or typographical errors that may have slipped past the professional editing process.

We are committed to ensuring that the final manuscript meets PLOS ONE's high standards for clarity and correctness of language. We appreciate your understanding and patience, and we look forward to any further suggestions that you may have to improve our manuscript.

5. 

(i) Comment: The paper has investigated an important and well-timed topic. Yet the paper lacks many elements.

Response: We sincerely appreciate your comprehensive feedback. We have made considerable efforts in this revised manuscript to address each of your comments and concerns in turn.

(ii) Comment: Abstract: It has not expressed the purpose of this research.

Response: We have revised our abstract to better reflect the purpose of our research. We have reduced the emphasis on research design and instead focused more on our research purpose of this research and the key findings.

Here is the rewritten abstract:

While the rapid digitalization in higher education, accelerated by the COVID-19 pandemic, has restructured the landscape of teaching and learning, a comprehensive understanding of its implications on students' academic outcomes across various academic disciplines remains unexplored. This study, therefore, aims to fill this gap by providing an in-depth examination of the effects of crisis-driven digitalization on student performance, specifically the shift to emergency remote education during the COVID-19 crisis. Leveraging a panel dataset encompassing 82,694 individual student course grades over a span of six years, we explore the effects of digitalization across nationalities, educational levels, genders, and crucially, academic disciplines. 

Our findings are threefold: (i) firstly, we note that crisis-driven digitalization significantly impacted students’ chances of passing a course and achieving higher course grades in comparison to the pre-crisis period. (ii) Secondly, we found the effect to be heterogeneous across disciplines. Notably, practical disciplines, such as nursing, experienced a negative impact from this sudden shift, in contrast to more theoretical disciplines such as business administration or mathematics, which saw a positive effect. (iii) Lastly, our results highlight significant variations in the impact based on educational levels and nationalities. Master’s students had a harder time adapting to the digital shift than their bachelor counterparts, while international students faced greater challenges in less international academic environments. These insights underscore the need for strategic interventions tailored to maximize the potential of digital learning across all disciplines and student demographics. The study aims to guide educators and policymakers in creating robust digital learning environments that promote equitable outcomes and enhance students' learning experiences in the digital age.

(iii) Comment: Keywords should be increased.

Response: We have updated our keywords to reflect the broader scope and important aspects of our research.

*Correction* � KEYWORDS: Digitalization; higher education; COVID-19; academic disciplines; academic outcomes; course outcomes; student performance; fixed effects; panel data analysis; natural experiment; online learning, heterogeneous effects; Swedish university; remote learning impact.

(iv) Comment: Introduction: The introduction lacks providing an explicit importance of a research problem.

Response: In the revised introduction, we have completely restructured our Introduction section, and we believe that the importance of our research problem, as well as the identified research gap, the aim, and the objectives of our study, are now more clearly defined. We thank the Reviewer 2 for pointing this out.

(v) Comment: The author neither makes a comprehensive literature review to give light to a conceptual framework or to develop a theoretical underpinning.

Response: In response to your feedback, we have expanded our literature review to better build a theoretical underpinning for our study. Amongst others, we included the following recommended paper: 

• Does online technology provide sustainable HE or aggravate diploma disease? Evidence from Bangladesh—a comparison of conditions before and during COVID-19- DOI: 10.1016/j.techsoc.2021.101677

(vi) Comment: Research design: Naming university name is not a right fashion. It breaks the research confidentiality issue. 

Response: We understand your concern. In the revised manuscript, we have anonymized the university name to maintain confidentiality.

(vii) Comment: Research design: The data collection phase is not explained and justified.

Response: Our analysis is based on administrative data collected by the university service. This data was hence already collected as part of the administrative system adopted at the university. This research project hence was provided the data and data additional data was not collected. We have carefully gone through the Data description part and think that the data is explained in a clear manner. If further revision is needed to this part we kindly ask the reviewer to be more specific on which parts that are not explained or justified. 

(viii) Comment: Results must be presented with coherence following a theme.

Response: We appreciate the feedback concerning the coherence of our results section. In response, we've undertaken a thorough revision to streamline the presentation of our findings. By restructuring the paper to align with a conventional format, we believe we've enhanced the clarity and thematic consistency of our results. The results are presented to directly address our stated research questions, ensuring a logical flow and comprehensive understanding. We trust that these modifications have improved the coherence and thematic continuity of our results section.

(ix) Comment: Theoretical implication is not expressed. And there is no convincing conclusion/ concluding remark.

Response: Thank you for pointing this lack of implication and concluding remark. We have expanded the implications of our findings and provided a more substantial and conclusive closing remark.

In paper text discussion: 

This paper contributes to the literature on digitalization by identifying the impacts of online teaching in various academic disciplines, avoiding selection bias. Our results highlight the importance of designing effective digital strategies that maximize the potential of online teaching and learning across all academic disciplines. Educators and institutions should consider the specific needs and challenges of different disciplines when implementing digital tools and pedagogies. Practical disciplines, for example, may require more hands-on and experiential learning approaches, even in an online environment. The variation in the impact of digitalization across educational levels and nationalities emphasizes the need for tailored support mechanisms. Master's students may require additional assistance and resources to adapt to online learning, while international students may benefit from targeted support to navigate challenges in less international academic environments. This customization calls for investments in competence development for educators to enhance their digital teaching skills. Educators need training and support to effectively engage students in online learning environments and overcome the disciplinary-and context-specific challenges identified in this study. The unique needs of different disciplines and student populations also suggest that a one-size-fits-all approach may not be suitable. Policies should be flexible. 

Finally, our findings point towards the importance of preparing students for digital learning experiences. As pointed out by previous literature, students need digital literacy skills and resources to effectively engage with online coursework. Providing training and support to ensure students are equipped with the necessary tools and knowledge to succeed in a digital learning environment is crucial. By considering these implications, stakeholders can work together to optimize digital learning experiences, promote equitable outcomes across disciplines and student groups, and ensure that digitalization leads to enhance students' learning experiences in the digital age.

(x) Comment: Language demands editing. Serious attentions are needed to be paid for technical issues such as citations, references and formatting.

Response: Thank you for your comment regarding the language and clarity of our manuscript. We want to clarify that before our initial submission, the manuscript had undergone a thorough proofreading and editing process by professionals, as none of the authors are native English speakers. We have taken your feedback into account and have made further revisions to address any language inconsistencies or ambiguities that might have remained. However, if there are specific areas you identified as needing improvement, we would greatly appreciate further details so we can address them directly.

Moreover, in our initial submission, we used Mendeley reference manager to manage and format our citations and references. However, in response to your feedback, we have thoroughly rechecked every citation and reference in the manuscript to ensure they adhere to the journal's style guide and are formatted correctly. If there are any specific issues you noted with our referencing, we would appreciate further details so we can address them directly.

We appreciate your constructive feedback and hope that these revisions adequately address your concerns. We look forward to your further comments and suggestions.

---

## [Editor Report · Decision Letter 1]

17 Oct 2023

Crisis-driven digitalization and academic success across disciplines.

PONE-D-23-00621R1

Dear Dr. Tinjić,

We’re pleased to inform you that your manuscript has been judged scientifically suitable for publication and will be formally accepted for publication once it meets all outstanding technical requirements.

Kind regards,

Mukhtiar Baig, Ph.D.

Academic Editor

PLOS ONE

---

## [Editor Report · Acceptance letter]

27 Oct 2023

PONE-D-23-00621R1 

Crisis-driven digitalization and academic success across disciplines. 

Dear Dr. Tinjić:

I'm pleased to inform you that your manuscript has been deemed suitable for publication in PLOS ONE. Congratulations! Your manuscript is now with our production department. 

Kind regards, 

on behalf of

Professor Mukhtiar Baig 

Academic Editor

PLOS ONE